

# Clarifying a trophic black box: stable isotope analysis reveals unexpected dietary variation in the Peruvian anchovy *Engraulis ringens*

Jessica Pizarro[1,2], Felipe Docmac[3,4] and Chris Harrod[3,4,5]

[1] Facultad de Recursos Naturales Renovables, Universidad Arturo Prat, Iquique, Chile
[2] Departamento de Ecología, Facultad de Ciencias, Universidad Autónoma de Madrid, Madrid, Spain
[3] Instituto de Ciencias Naturales Alexander von Humboldt, Universidad de Antofagasta, Antofagasta, Chile
[4] Universidad de Antofagasta Stable Isotope Facility (UASIF), Instituto Antofagasta, Universidad de Antofagasta, Antofagasta, Chile
[5] Núcleo Milenio INVASAL, Concepción, Chile

Corresponding author
Chris Harrod, chris@harrodlab.net

## ABSTRACT

**Background**. Small fishes play fundamental roles in pelagic ecosystems, channelling energy and nutrients from primary producers to higher trophic levels. They support globally important fisheries in eastern boundary current ecosystems like the Humboldt Current System (HCS) of the SE Pacific (Chile and Peru), where fish catches are the highest in the world (per unit area). This production is associated with coastal upwelling where fisheries target small pelagic fishes including the Peruvian anchovy (*Engraulis ringens*). The elevated biomass attained by small pelagics is thought to reflect their low trophic position in short/simple food chains. Despite their global importance, large gaps exist in our understanding of the basic ecology of these resources. For instance, there is an ongoing debate regarding the relative importance of phytoplankton versus animal prey in anchovy diet, and ecosystem models typically assign them a trophic position (TP) of ~2, assuming they largely consume phytoplankton. Recent work based on both relative energetic content and stable isotope analysis (SIA) suggests that this value is too low, with $\delta^{15}N$ values indicating that anchovy TP is ca. 3.5 in the Peruvian HCS.
**Methods**. We characterised the trophic ecology of adult anchovies ($n = 30$), their putative prey and carnivorous jack mackerel ($n = 20$) captured from N Chile. SIA ($\delta^{13}C$ and $\delta^{15}N$) was used to estimate the relative contribution of different putative prey resources. $\delta^{15}N$ was used to estimate population level trophic position.
**Results**. Anchovies showed little variability in $\delta^{13}C$ (−18.7 to −16.1‰) but varied greatly in $\delta^{15}N$ (13.8 to 22.8‰)—individuals formed two groups with low or high $\delta^{15}N$ values. When considered as a single group, mixing models indicated that anchovy diet was largely composed of zooplankton (median contribution: 95% credibility limits), with major contributions of crustacean larvae (0.61: 0.37–0.77) and anchovy (preflexion) larvae (0.15: 0.02–0.34), and the assimilation of phytoplankton was negligible (0.05: 0–0.22). The modal (95% credibility limits) estimate of TP for the pooled anchovy sample was 3.23 (2.93–3.58), overlapping with recent SIA-based estimates from Peru. When the two $\delta^{15}N$ groups were analysed separately, our results indicate that the lower $\delta^{15}N$ group largely assimilated materials from crustacean larvae (0.73: 0.42–0.88), with a TP of 2.91 (2.62–3.23). Mixing models suggested high $\delta^{15}N$

anchovies were cannibalistic, consuming anchovy preflexion larvae (0.55: 0.11–0.74). A carnivorous trophic niche was supported by high TP (3.79: 3.48–4.16), mirroring that of carnivorous juvenile jack mackerel (*Trachurus murphyi*; 3.80: 3.51–4.14). Our results support recent conclusions regarding high TP values of anchovy from Peru and reveal new insights into their trophic behaviour. These results also highlight the existence of cryptic trophic complexity and ecosystem function in pelagic food webs, classically considered as simple.

## INTRODUCTION

Boundary current ecosystems such as the Humboldt Current System (HCS) are characterised by high biological productivity driven by coastal upwelling of cold sub-surface nutrient-rich waters (*Chavez & Messié, 2009*). Food chains in such ecosystems have been typically considered as simple and short (*Ryther, 1969*), with high efficiency of trophic transfer between trophic levels. Here, phytoplankton form the base of the food web, and are consumed by zooplankton and small-bodied pelagic fishes such as the Peruvian anchovy *Engraulis ringens* Jenyns, 1842 (from hereon in, anchovy). The small pelagic fish assemblage is typically dominated by one or a few species (*Bakun, 1996*), which due to their sheer abundance and biomass can exercise control over the trophic dynamics of the whole ecosystem (*Cury et al., 2000*).

Biological production in the HCS (*Herrera & Escribano, 2006*) is such that it supports the capture of more fish per unit area than any other environment in the world (*Chavez et al., 2008*). Indeed, production in the HCS is considered anomalous, even among eastern current systems (*Bakun & Weeks, 2008*; *Chavez & Messié, 2009*). Industrial fisheries in Peru and Chile target anchovies and other pelagic fishes such as jack mackerel *Trachurus murphyi* and sardines (*Sardinops sagax*). The anchovy stock in the region is the most heavily exploited fish worldwide: although annual catches vary considerably reflecting the marked effects of ENSO on stock size, between 1990 and 2015, on average (median $\pm$ IQR) 7,419,295 $\pm$ 3,329,637 tonnes of anchovies were removed per year (*FAO, 2018*).

Although anchovy (and other small pelagics) support globally important fisheries (*Chavez & Messié, 2009*; *Kämpf & Chapman, 2016*), and play integral roles in structuring upwelling ecosystems (*Cury et al., 2000*), many basic aspects of their ecology are understudied. For instance, information on their trophic ecology is limited, and likely constrains our understanding of their role in the HCS, and the utility of ecosystem model outputs. Studies of anchovy diet based on stomach content analysis (SCA) of juveniles and adults typically report that stomach contents are dominated (in numerical terms) by phytoplankton (*Espinoza & Bertrand, 2008*; *Medina et al., 2015*; *Ryther, 1969*), and anchovies have been classically considered as phytoplanktivores. This low trophic position

has been proposed as an explanation for their great abundance in the HCS, and the ecosystem's anomalous capacity to support such high production of pelagic fishes.

Most studies of anchovy diet in the literature are based on counts of individual stomach contents, which is known to bias conclusions when prey of markedly different sizes are consumed (*Hyslop, 1980*). Not surprisingly, microscopic food items such as diatoms can be extremely abundant, leading them to dominate relative estimates of prey importance (up to 98%: *Whitehead, Nelson & Wongratana, 1998*), even though zooplankton are commonly reported from anchovy stomach contents. Given the putative importance of phytoplankton to their diet, many researchers have assumed that anchovy have a trophic position of ca. 2–2.5 (*Guénette, Christensen & Pauly, 2008*; *Pauly et al., 1998*). These values are commonly used in ecosystem models such as Ecopath, and anchovies are considered a classic low-trophic level species (*Smith et al., 2011*). This estimate of anchovy TP has played an important role in one of the largest controversies in modern fisheries science—the issue of global fishing down the food web (*Pauly et al., 1998*; *Pauly, Froese & Palomares, 2000*). The global importance of the HCS anchovy fishery is such that if its statistics are included, it skews global estimates of the mean trophic level of catch downward.

More recently however, there has been an important reassessment of anchovy diet. *Espinoza & Bertrand (2008)*, rather than counting stomach contents, focused on the energetic content of prey, and clearly showed that zooplankton (mainly copepods and euphausiids) were the principle source of energy to anchovies. Beyond highlighting a key issue of using counts of stomach contents to estimate TP, their results had major implications regarding our understanding of how this globally important marine ecosystem functions (e.g., *Ballón et al., 2011*). Given the biomass of anchovies in the HCS, and their dominant role in the food web, changes in our understanding of their putative diet has subsequent impacts on how we interpret the movement of energy and nutrients through the food web, with consequences for qualitative and quantitative models of HCS function and resource management.

SCA has long been the gold standard for assessing what fish consume (*Hynes, 1950*; *Hyslop, 1980*), but clearly can bias our understanding of how fish direct the flow of materials through a food web (*Espinoza & Bertrand, 2008*; *Hyslop, 1980*). An alternative to the snapshot of recently ingested prey provided by SCA, is to take a biochemical approach to assessing diet such as stable isotope analysis (SIA) or fatty acid analysis (*Nielsen et al., 2018*). The advantages of SIA in particular, are that the technique provides information on prey assimilation over longer temporal scales than SCA (weeks–months), with the period depending on the tissue sampled (*Thomas & Crowther, 2015*). By combining analysis of carbon and nitrogen stable isotope ratios of consumers and their putative prey, it is possible to characterise the source of energy and nutrients assimilated by a consumer (*Parnell et al., 2013*). Furthermore, if isotope values are available for both the consumer and the base of the food web (*Vander Zanden & Rasmussen, 1996*), it is possible to estimate the long term trophic position at which a consumer feeds (*Quezada-Romegialli et al., 2018*).

Recent work in central Chile and Peru using SIA has provided further evidence that previous assumptions regarding anchovy trophic position were wrong. *Hückstadt, Rojas & Antezana (2007)* compared POM and anchovy $\delta^{15}$N values from central Chile and

estimated anchovy TP as 3.6. In a wide-ranging recent study from the northern HCS, *Espinoza et al. (2017)* compared anchovy (and other taxa) $\delta^{15}$N values relative to copepod $\delta^{15}$N to estimate TP. They estimated that anchovy TP ranged between 3.4 and 3.7. Both of these studies provide evidence that anchovy TP is at least a trophic level above values typically used for the species in trophic models (*Guénette, Christensen & Pauly, 2008*; *Pauly et al., 1998*), and challenge the concept of anchovies as a low trophic level fish (*Smith et al., 2011*). *Espinoza et al.*'s (*2017*) estimates for anchovy TP are important as they support the authors' previous findings from SCA indicating that phytoplankton, although abundant numerically in stomachs, did not make a major contribution to assimilated diet. However, their and *Hückstadt, Rojas & Antezana*'s (*2007*) estimates were based on a simple model which does not include isotopic variation in trophic discrimination or in the baseline itself (see *Quezada-Romegialli et al., 2018* for a description of the issue). The latter point is particularly important in the *Espinoza et al. (2017)* case, as the use of copepods as a baseline likely introduces considerable error compared to e.g., the use of a primary producer, given the range of trophic strategies displayed by pelagic marine copepods (*Giesecke & González, 2004*) and the uncertainties in allocating a TP for mixed samples of copepods.

It is becoming increasingly apparent that food webs associated with upwelling ecosystems can be more complex and dynamic than previously thought (*Docmac et al., 2017*; *Espinoza & Bertrand, 2008*). As such, there is a need for improved understanding of how these systems function, in order to inform and update existing trophic models used to explain the flow of energy and nutrients, as well as to allow an informed management of a globally important fishery. Here, we examine the trophic ecology of adult anchovies from northern Chile using stable isotope ratios of anchovies and their putative prey to estimate the relative role of phytoplankton and other prey, and to provide robust estimates of anchovy trophic position. Given the debate over the trophic ecology of anchovy, we compared their stable isotope values with that of juvenile jack mackerel, a known pelagic carnivore (*Alegre et al., 2015*; *Orrego & Mendo, 2015*).

## MATERIALS & METHODS

### Study area

Samples were collected between 20°30′S and 21°30′S off the coast of northern Chile (Fig. 1) during the Austral winter of 2008. This area is characterized for having persistent winds that permit year-round upwelling (although upwelling often strengthens during Spring–Summer), generating the intrusion of cold nutrient rich sub-surface waters along the shore (*Thiel et al., 2007*).

During the study period, neutral environmental conditions were present (i.e., non-ENSO). Sea surface temperatures ranged between 15.2 and 17.5 °C, salinities varied between 34.6 and 34.9 and surface dissolved oxygen concentrations between 4.2 and 6.7 ml l$^{-1}$. In general terms, the study area was characterized by a low-magnitude permanent upwelling, more marked around a latitude of 21°10′S (*Fuenzalida et al., 2009*).
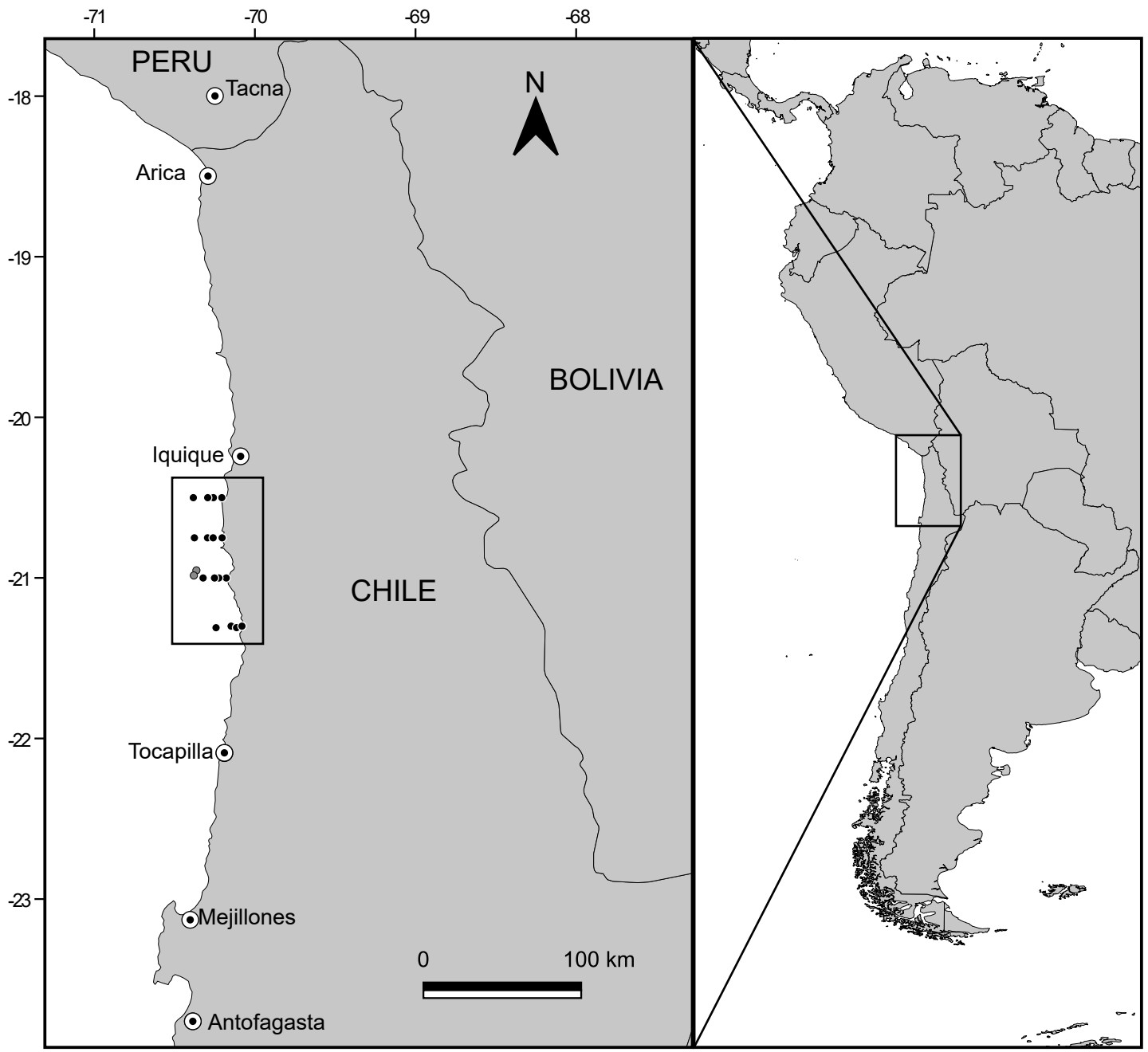

**Figure 1  Study area and position of sampling stations off the coast of N Chile.** Dark markers represent the location where putative prey were captured, while the light grey markers show the capture location of the anchovy and jack mackerel used in the study.

## Sample collection

Fish were captured (Fig. 1) at 20°57′S–70°22′W (adult anchovies) and 20°59′S–70°23′W (juvenile jack mackerel to allow comparisons with a known carnivorous fish) by the fishing vessel Atacama V, using a commercial purse seine (mesh 15 mm). Once captured, fish were frozen at −20 °C until further analysis. Permission to undertake field sampling was

provided by the Chilean Subsecretaría de Pesca through Resolución Exenta No. 2205 (21 August 2008).

Samples of putative planktonic prey (phytoplankton and zooplankton) were collected during daylight hours following a grid design along four longitudinal transects (latitudinal distance between transects 15 nautical miles), located perpendicular to the N Chilean coast (Fig. 1). Each longitudinal transect included four stations, located at 1, 3, 5 and 10 nautical miles from the coast, resulting in a total of 16 sampling sites. Phytoplankton was collected via vertical tows (from 50 m) of a phytoplankton net (20 $\mu$m mesh size, 20 cm opening). Zooplankton was collected through vertical tows (max. depth 100 m, but varied with sampling station depth) of a WP-2 net (300 $\mu$m mesh size, 50 cm opening). Samples were immediately frozen at $-20\,°C$, and following transfer to the laboratory placed in a $-80\,°C$ freezer.

## Laboratory analysis

Once defrosted, total length (TL: $\pm 1$ mm) and blotted wet mass ($\pm 0.1$ g) were estimated for anchovy ($n = 30$) and jack mackerel ($n = 20$). Stomachs were removed for analysis of stomach contents, but on inspection, a large majority of anchovy stomach contents were found to be in an advanced stage of digestion, limiting the utility of SCA in this case and we do not consider these data further in this study.

A sample of dorsal muscle was excised and treated with chloroform:methanol (2:1) to extract lipids (*Bligh & Dyer, 1959*). Samples were then oven dried (60 °C for 48 h) and homogenised prior to analysis of carbon and nitrogen stable isotope values.

Prior to SIA, phytoplankton samples were pre-filtered through a 200 $\mu$m sieve to remove zooplankton and large detritus. Zooplankton were sorted into different components: anchovy eggs and larvae, unidentified fish eggs, amphipods, appendicularians, chaetognaths, copepods, crustacean larvae, euphausiids, ostracods, pelagic polychaetes, salpidae and siphonophora. Each of these groups were pooled to obtain sufficient mass for SIA.

Phytoplankton and zooplankton samples were washed with milli-Q water and then filtered through pre-combusted (450 °C × 4 h) GF/F filters (Whatmann, 0.7 $\mu$m pore size, 47 mm diameter). Filters were subsequently oven dried (60 °C for 48 h), then acidified by directly applying HCl (1N) for 24 h to remove inorganic carbon and re-dried (*Carabel et al., 2006*). Zooplankton samples were then scraped off filters prior to homogenisation. Fish and zooplankton samples were homogenised with an agate mortar and pestle and weighed into tin capsules (sample mass $\sim$0.5 mg). Phytoplankton samples were run from sections of GF/F filters.

Analysis of $\delta^{13}C$ and $\delta^{15}N$ were conducted at the Colorado Plateau Stable Isotope Laboratory in Northern Arizona University (USA) using a Costech ECS4010 elemental analyser coupled to a Delta Plus Advantage isotope ratio mass spectrometer in continuous flow mode via a ConFlo III interface. Isotopic abundances are expressed in $\delta$ notation (‰), using the formula $\delta^{13}C$ or $\delta^{15}N = [(R_{sample} - R_{standard})/R_{standard}] \times 10^3$; where: $R_{sample}$ is $^{13}C/^{12}C$ or $^{15}N/^{14}N$, $R_{standard}$ is Vienna Pee Dee Belemnite for $\delta^{13}C$ and atmospheric nitrogen for $\delta^{15}N$. NIST 1547 (peach leaves) were used as an internal laboratory working

standard to check on measurement reproducibility throughout each run (analytical precision of $\leq 0.1‰$ for $\delta^{13}C$ and $\leq 0.2‰$ for $\delta^{15}N$). A number of peach leaf standards that varied in mass (from 0.5 to 6 mg) were also included at the end of the run to check on linearity. Analytical data were normalized to international standards using calibrated reference standards with known $\delta^{13}C$ (IAEA CH6 & IAEA CH7) or $\delta^{15}N$ (IAEA N1 & IAEA N2) values.

## Data analysis

Data were plotted in the form of an $\delta^{15}N$-$\delta^{13}C$ scatterplot to provide a visual representation of the distribution of isotopic values for the different taxa examined, and to examine putative differences between anchovy and jack mackerel. Individual data were plotted for fish, while putative prey were plotted as means $\pm$ SD. Size differences between the fish species were examined using the Welch's $t$-test. We examined potential relationships between fish isotope values and their individual body mass using Spearman's rank correlation coefficient. Both Welch's t-tests and correlations were conducted in SYSTAT 13.1 (SYSTAT, Richmond, CA, USA). Unless otherwise reported, summary statistics reflect mean $\pm$ 1 SD.

In order to examine whether the two fish species differed isotopically, we compared $\delta^{15}N$-$\delta^{13}C$ centroids of anchovies and jack mackerel using a permutation-based multivariate analysis of variance (PERMANOVA: $n_{permutations} = 9,999$), based on a Euclidean similarity matrix of untransformed $\delta^{15}N$-$\delta^{13}C$ data (*Anderson, Gorley & Clarke, 2008*). Due to an apparent sub-structuring within the anchovies based on $\delta^{15}N$ values (see below), we also tested for differences between the two observed groups of anchovies using a similar approach. PERMANOVA analyses were conducted in PRIMER with PERMANOVA 7.0.13 (*Anderson, Gorley & Clarke, 2008*; *Clarke & Gorley, 2015*).

We used the R-based (version 3.5.0; *R Core Team, 2018*) Bayesian mixing model SIMMR to estimate the relative contribution of seven different key putative prey groups to anchovy and jack mackerel assimilated diet (*Parnell, 2016*). These groups were selected reflecting their abundance in zooplankton hauls and literature descriptions of anchovy diet (*Espinoza & Bertrand, 2008*; *Medina et al., 2015*). SIMMR was run in RStudio (version 1.1.447; *RStudio Team, 2016*) using default settings for the number of iterations, burn in and Markov chain Monte Carlo (MCMC) chains. Putative prey were grouped *a priori* as anchovy eggs, anchovy larvae, copepods, crustacean larvae, euphausiids, unidentified fish eggs and phytoplankton. We used mean $\pm$ SD trophic discrimination factors from *Post (2002)* ($\Delta^{13}C = 0.4 \pm 1.3‰$; $\Delta^{15}N = 3.4 \pm 1.0‰$), and included information on prey elemental concentrations.

We estimated TP of anchovy and jack mackerel in R using the Bayesian package tRophicPosition (*Quezada-Romegialli et al., 2018*), using the *Onebaseline* model. Briefly, the tRophicPosition model includes isotopic variation in the baseline indictor, the consumer and in the trophic discrimination in order to provide a robust estimation of consumer TP at the population level. We included the phytoplankton $\delta^{15}N$ data to provide the baseline ($\lambda = 1$), and assumed a trophic discrimination factor ($\Delta^{15}N$) of $3.4 \pm 1‰$ (*Post, 2002*).
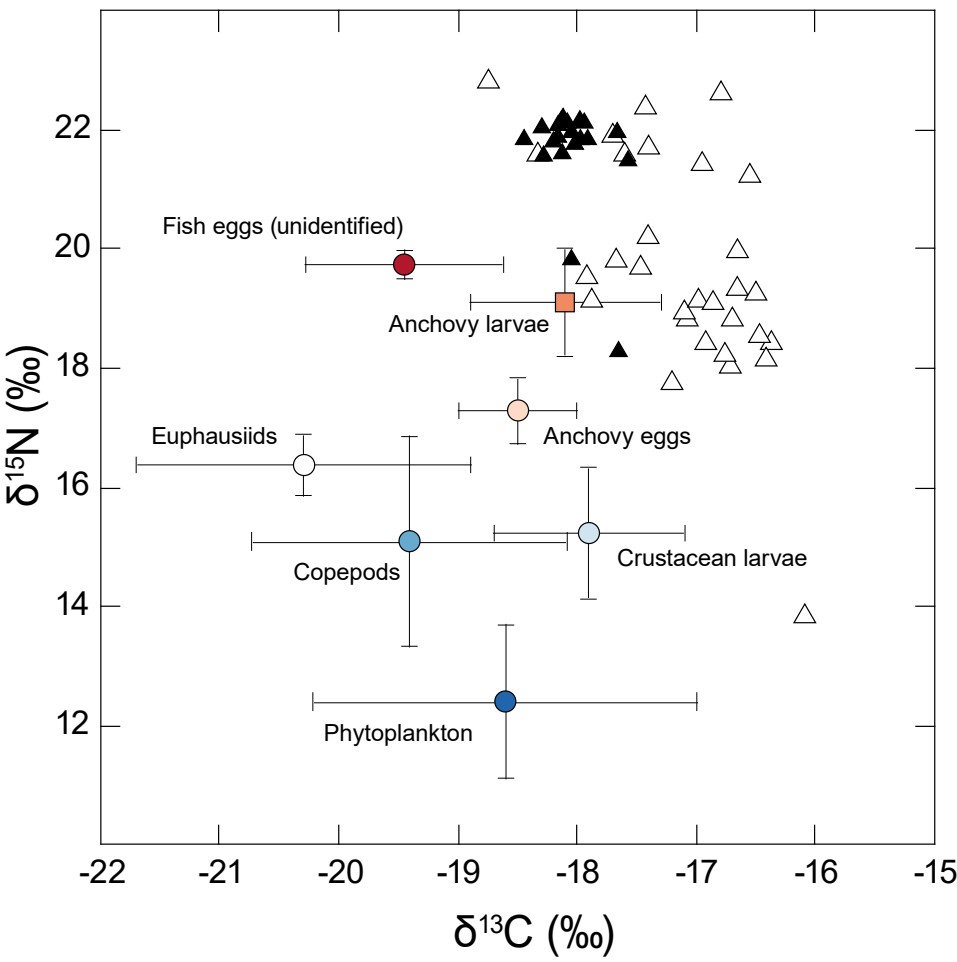

**Figure 2** **Stable isotope $\delta^{15}$N - $\delta^{13}$C biplot showing individual values for adult anchovies (open triangles) and juvenile jack mackerel (filled triangles) captured during the current study—putative prey values are shown as mean (±SD).** Note the considerable inter-individual differences in $\delta^{15}$N shown by anchovies. Fish $\delta^{13}$C values reflect lipid-extracted tissues, while putative prey were not treated, as consumers assimilate carbon from different biochemical compounds in their prey including lipids.

## RESULTS

Isotopic values for the different taxa analysed are shown in Fig. 2. Phytoplankton had a mean $\delta^{13}$C value of $-18.6‰$ but showed considerable variation between sampling locations (SD = 1.6‰). Phytoplankton mean $\delta^{15}$N was 12.4 ($\pm$ 1.3‰), reflecting the influence of $^{15}$N-enriched upwelling-derived NO$_3$, and apparently formed the basal resource for consumers from overlying trophic levels. Carbon stable isotope values for the different putative zooplankton prey classes (anchovy eggs and larvae, unidentified fish eggs, copepods, crustacean larvae, euphausiids) ranged between $-20.3$ and $-17.9‰$ for $\delta^{13}$C and 15.1 to 19.7‰ for $\delta^{15}$N.

Carbon and nitrogen stable isotope values were estimated from 30 adult anchovies (28 male, 2 female). Anchovy TL varied between 13.7 and 16.1 cm (mean TL = 14.8 ± 0.6 cm),

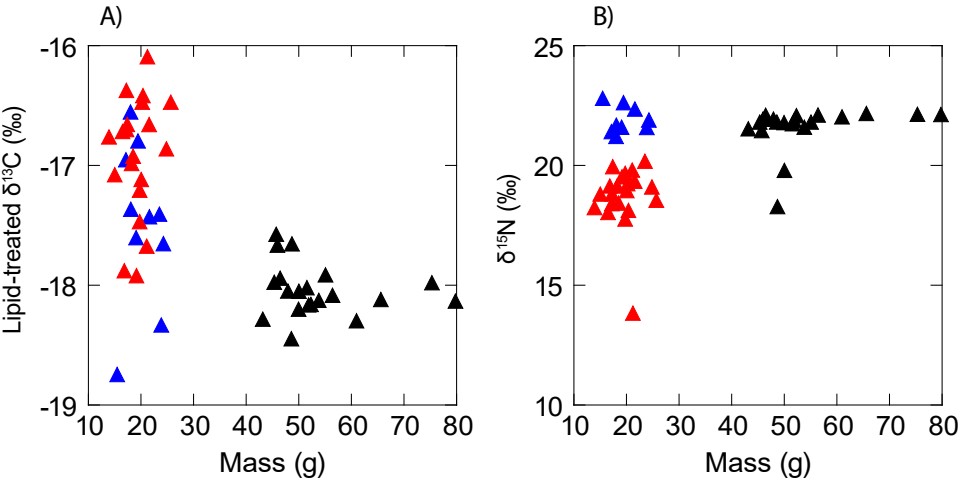

**Figure 3** Scatterplot showing a lack of any obvious relationship between (A) $\delta^{13}$C and (B) $\delta^{15}$N and fish mass in anchovy (low $\delta^{15}$N = red triangles; high $\delta^{15}$N = blue triangles) and juvenile jack mackerel (black triangles).

and blotted wet mass between 13.9 and 25.7 g (19.5 ± 2.9 g). Juvenile jack mackerel (one male, one female, 18 indeterminate) TL varied between 18.5 and 22.2 cm (19.5 ± 1.1 cm), and mass between 43.2 and 79.8 g (53.7 ± 9.8 g).

Anchovy lipid-free $\delta^{13}$C values varied between −18.7 and −16.1‰, with a mean of −17.1 ± 0.6‰. Variation in $\delta^{15}$N values was more marked, with individual anchovies ranging between 13.8 and 22.8‰, and a mean $\delta^{15}$N of 19.7 (±1.9)‰. Examination of anchovy $\delta^{15}$N values revealed the presence of two apparent groups differing in their $\delta^{15}$N values: one group was relatively $^{15}$N enriched (21.6 ± 0.9‰), and the second was made up of $^{15}$N-depleted individuals (18.5 ± 1.3‰).

Jack mackerel lipid-free $\delta^{13}$C values ranged between −18.4 and −17.6‰ (mean $\delta^{13}$C −18.0 ± 0.2‰), while their $\delta^{15}$N values ranged between 18.3 and 22.2‰, with a mean $\delta^{15}$N of 21.6 (± 0.9)‰, apparently similar to that of the high $\delta^{15}$N group of anchovies.

There was no evidence of correlation between the isotopic composition and mass (Fig. 3) of anchovy ($\delta^{13}$C: $R_s = -0.02$, $P = 0.94$; $\delta^{15}$N: $R_s = 0.11$, $P = 0.56$). Jack mackerel $\delta^{13}$C was similar across the size range examined ($R_s = -0.27$, $P = 0.27$) but showed a positive relationship between mass and $\delta^{15}$N ($\delta^{15}$N: $R_s = 0.55$, $P = 0.01$). Anchovies were significantly smaller than jack mackerel (Welch's $t$ test of mass: $t_{28.1} = -15.1$, $P < 0.001$), but there was no evidence for any size difference between the two anchovy groups ($t_{17.2} = 0.70$, $P = 0.493$).

## Isotopic differences

PERMANOVA indicated that $\delta^{13}$C-$\delta^{15}$N centroids differed between the anchovy and jack mackerel (Pseudo-$F_{1,49} = 20.6$, $P = 0.0001$). When anchovies are considered as two different groups (high and low $\delta^{15}$N), there was again strong evidence for isotopic differences between the three groups of fish compared: low $\delta^{15}$N adult anchovies, high $\delta^{15}$N

adult anchovies and juvenile jack mackerel (Pseudo-$F_{2,49} = 45.2$, $P = 0.0001$). Pairwise tests however revealed significant overlap between juvenile jack mackerel and the individuals in the high $\delta^{15}$N anchovy group ($P = 0.11$). Anchovies from the low $\delta^{15}$N group were isotopically different from both ($P = 0.0001$ in both cases).

A univariate PERMANOVA showed significant differences in $\delta^{13}$C between groups (Pseudo-$F_{2,49} = 30.3$, $P = 0.0001$), with higher $\delta^{15}$N anchovies being ca. 0.6‰ depleted in $^{13}$C compared to individuals from the lower $\delta^{15}$N group. A similar analysis showed that the three groups of fish differed in their $\delta^{15}$N values (Pseudo-$F_{2,49} = 47.9$, $P = 0.0001$) and that the group of anchovies lower in $\delta^{15}$N was $^{15}$N depleted by ca. 3‰ (close to one trophic level given a TDF of 3.4‰) compared to the higher $\delta^{15}$N anchovies and the jack mackerel (Pairwise $P = 0.0001$). $\delta^{15}$N values overlapped between the high $\delta^{15}$N anchovy group and the jack mackerel ($P = 0.74$), suggesting a similar feeding mode.

## Mixing models

Mixing models results (Fig. 4, Table 1) indicate that all fish examined were carnivorous, with little evidence for large-scale contributions by phytoplankton. When considered as a single group, anchovies largely assimilated carbon and nitrogen from crustacean larvae (median estimate = 61%) and anchovy larvae (15%), with phytoplankton making a minor contribution (5%). When considered separately, the anchovy group with high $\delta^{15}$N values had a large contribution of anchovy larvae in their diet (55%), while the low $\delta^{15}$N anchovy group were estimated to have assimilated the majority of their carbon and nitrogen from crustacean larvae (73%). Mixing model results indicated that anchovy larvae (39%), unidentified fish eggs (19%) and anchovy eggs (15%) all made major contributions to the assimilated diet of jack mackerel. Comparison between the mixing model results for the high $\delta^{15}$N group of anchovies and jack mackerel indicated that they had broadly similar foraging modes.

## Trophic position

Bayesian estimates of TP supported the mixing model results, with credibility limits for all fish including TP 3 (Fig. 5). Modal (95% credibility limits) estimated TP for the pooled anchovy sample was 3.23 (2.93–3.58). The low $\delta^{15}$N anchovy group had a lower estimated TP (2.91: 2.62–3.23). Estimates of TP for the high $\delta^{15}$N anchovy group (3.79: 3.48–4.16) and jack mackerel (3.80: 3.51–4.14) were highly similar.

## DISCUSSION

Our study aimed to further our understanding of the trophic ecology of the Peruvian anchovy—a fish that supports the most productive fishery in the world. Existing data on anchovy diet, like that of other pelagic fishes, is largely derived from the analysis of stomach contents, and has generally concluded that anchovy production in the study region is fuelled by the consumption of phytoplankton (*Medina et al., 2015*) and copepods (*Castillo et al., 2002*; *Castillo et al., 2011*). Our results, based on stable isotope values, indicate that the contribution of phytoplankton to anchovy somatic tissues is minimal and that zooplankton represents the main food source assimilated by anchovies. Our results

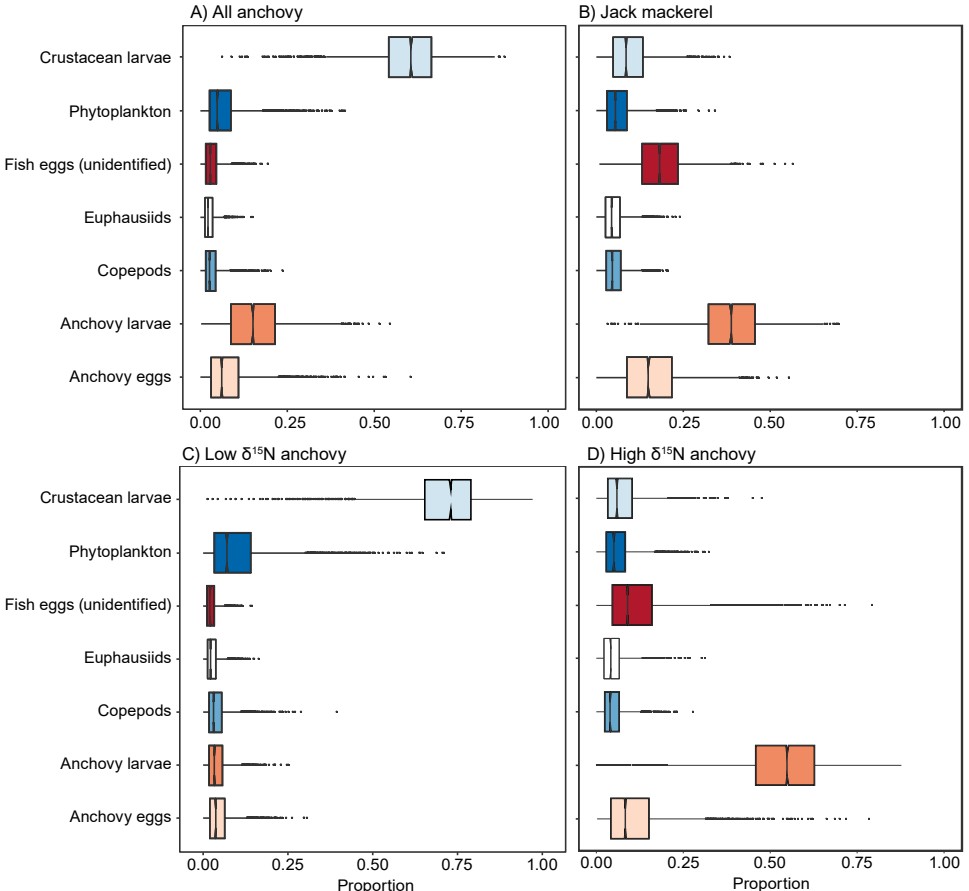

**Figure 4 Visual representation of SIMMR mixing model results.** Boxplots showing the distribution of posterior estimates of contribution of the different putative prey to assimilated diet to: (A) all anchovy, (B) jack mackerel, (C) low $\delta^{15}$N anchovies, and D) high $\delta^{15}$N anchovies.

follow those of Espinoza and colleagues (*Espinoza & Bertrand, 2008*; *Espinoza et al., 2017*), who have elegantly demonstrated that a reliance on stomach content data has resulted in a significant misinterpretation of anchovy trophic ecology and by extension, the food web of the HCS. Beyond our reaffirmation of *Espinoza et al.*'s observations (*2017*) regarding the trophic position of anchovy, we have also identified several previously unknown features of the pelagic food web of the Chilean HCS, suggesting that the trophic ecology of the Peruvian anchovy, and ecosystem function in the HCS are both more complex than is typically considered.

The HCS has several characteristic features associated with elevated fisheries production. Using stable isotopes of anchovies, jack mackerel and their putative prey, as well as the key pelagic primary producer, phytoplankton, we were able to highlight the influence of upwelling throughout the food chain in the Chilean HCS. Phytoplankton and higher trophic levels were naturally-labelled with heavy nitrogen associated with upwelling (*Casciotti, 2016*; *Docmac et al., 2017*; *Reddin et al., 2015*), and results indicate that phytoplankton was the likely carbon source fuelling upper trophic levels in the HCS. We used phytoplankton

**Table 1** Summary statistics (median (credibility limits)) for estimated contribution of different putative prey to the assimilated diet of anchovy (all anchovy combined, low $\delta^{15}$N and high $\delta^{15}$N groups) and jack mackerel estimated using SIMMR.

| Putative prey | Estimated proportional contribution to diet | | | |
|---|---|---|---|---|
| | All anchovy | High $\delta^{15}$N anchovy | Low $\delta^{15}$N anchovy | Jack mackerel |
| Anchovy eggs | 0.06 (0.01–0.26) | 0.08 (0.01–0.36) | 0.04 (0.01–0.15) | 0.15 (0.02–0.36) |
| Anchovy larvae | 0.15 (0.02–0.34) | 0.55 (0.11–0.74) | 0.03 (0.01–0.13) | 0.39 (0.18–0.59) |
| Copepods | 0.03 (0.01–0.1) | 0.04 (0.01–0.14 | 0.03 (0.01–0.13) | 0.05 (0.01–0.13) |
| Crustacean larvae | 0.61 (0.37–0.77) | 0.06 (0.01–0.23) | 0.73 (0.42–0.88) | 0.09 (0.01–0.23) |
| Euphausiids | 0.02 (0.001–0.08) | 0.04 (0.01–0.15) | 0.02 (0.004–0.08) | 0.05 (0.01–0.14) |
| Unidentified fish eggs | 0.03 (0.001–0.10) | 0.09 (0.01–0.44) | 0.02 (0.004 –0.07) | 0.19 (0.05–0.34) |
| Phytoplankton | 0.05 (0.001–0.22) | 0.05 (0.01–0.17) | 0.07 (0.01–0.36) | 0.06 (0.01–0.18) |

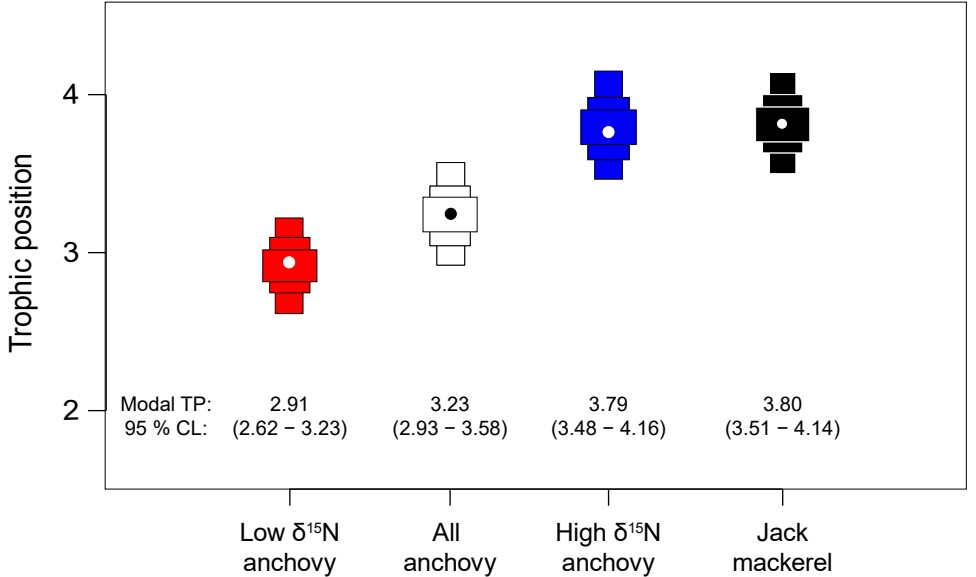

**Figure 5** Estimates of trophic position for anchovies and jack mackerel. Bars show the modal TP (circle) ±95% credibility limits, calculated relative to a phytoplankton baseline (TP1) using tRophicPosition.

collected across the survey area both as a putative food source in mixing models and as a baseline for the estimation of trophic position. Phytoplankton $\delta^{13}$C values varied by ca. 3‰ between locations, likely reflecting local differences in primary production rates due to variation in upwelling intensity, $CO_2$ concentration, phytoplankton growth rate etc., (*Magozzi et al., 2017*) and possibly community composition (*Fry & Wainright, 1991*). It is notable that several taxa examined here were notably $^{15}$N-enriched relative to their

counterparts in the northern HCS as reported by *Espinoza et al. (2017)* (e.g., copepods by 7.5‰, euphausiids by 7.6‰, anchovies by 7.6‰ and jack mackerel by 4.1‰). This likely reflects regional differences in upwelling intensity.

Anchovies have typically been considered to have homogeneous diets and therefore trophic position (*Medina et al., 2015*). Using SIA, however, we have shown the existence of considerable inter-individual differences in $\delta^{15}N$, a robust indicator of trophic position (*Post, 2002*). The differences between the two approaches presumably reflects an overestimation on the contribution of phytoplankton due to using count data (*Espinoza & Bertrand, 2008*) as well as the different timescales associated with SCA (hours) compared with SIA (months) (*Nielsen et al., 2018*; *Thomas & Crowther, 2015*), but highlights the power of SIA to identify cryptic individual variation (*Bolnick et al., 2002*; *Harrod et al., 2005*).

The observed variability in $\delta^{15}N$ values for adult anchovies very likely reflects differences in individual feeding strategies. We identified two trophic groups, one relatively enriched in $^{15}N$, associated with the consumption of anchovy larvae, and another, depleted in $^{15}N$ that largely consumed crustacean larvae. Anchovies from the first group had similar $\delta^{15}N$ values to juvenile jack mackerel, implying that they fed at a similar TP: however they differed in $\delta^{13}C$ values, likely indicating resource segregation. Anchovies in the lower $\delta^{15}N$ group had a modal (95% credibility interval) estimated TP of 2.91 (2.62–3.23) indicating some omnivory, but a largely carnivorous diet. Anchovies in the higher $\delta^{15}N$ group had a modal value of 3.79 (3.48–4.16) which overlapped completely with carnivorous jack mackerel (3.80 (3.51–4.14)), indicating that they are functionally similar. Support for a high TP for some anchovy is provided by *Espinoza et al. (2017)* who also showed overlap between jack mackerel $\delta^{15}N$ values and anchovies with high $\delta^{15}N$ values from southern Peruvian latitudes.

When the two anchovy groups were pooled, estimated TP (3.23 (2.93–3.58)) was ca. 1 TP higher than values estimated from stomach contents data and widely used in modelling studies (*Guénette, Christensen & Pauly, 2008*; *Pauly et al., 1998*). However, these pooled estimates effectively overlap with values from other studies using stable isotope analysis, such as *Hückstadt, Rojas & Antezana (2007)* working in central Chile (anchovy TP = 3.6) and *Espinoza et al. (2017)* from the Peruvian part of the HCS (3.4–3.7).

Our sample of anchovy was relatively small ($n = 30$), with individuals falling into two broad groups based on $\delta^{15}N$ values. It is unknown whether the patterns in inter-individual variability in anchovy $\delta^{15}N$ values we have shown are typical of the species, or even the genus. However, in their Fig. 5 *Espinoza et al. (2017)* show both a similar range of $\delta^{15}N$ values (12–20‰), as well as possible evidence for different $\delta^{15}N$ groups from anchovies from latitudes ca. 300–350 km north of where our samples were collected. It is possible that further intensive sampling may reveal that anchovy feed along a $\delta^{15}N$ continuum rather than in discrete groups as we suggest. It is likely that this is worthy of further study in other populations and *Engraulis* species.

An alternative explanation for the existence of two groups of anchovy differing in $\delta^{15}N$ (which we associate with differences in TP) is that the sample included fish from different geographical locations, where baseline $\delta^{15}N$ differed, but that had actually fed

at similar TPs. Baseline $\delta^{15}$N varies considerably along the Pacific coast of Chile and Peru, reflecting differences in the intensity of upwelling and the associated denitrification process. De Pol-Holz and colleagues (*2009*) reported a marked South–North gradient of $^{15}$N enrichment in marine sediments along the Chilean coast. We have shown elsewhere (*Docmac et al., 2017*) that consumer $\delta^{15}$N increases along a South-North gradient in the study region, and that $\delta^{13}$C co-varies spatially across the same scale. Given that the two putative groups of anchovy had similar $\delta^{13}$C values, we feel it is unlikely that they had different origins, and that we are confident in our identification of the two groups feeding at different trophic positions.

There was no evidence for any relationships between individual body mass and $\delta^{13}$C or $\delta^{15}$N in anchovy, as seen in the northern HCS by *Espinoza et al. (2017)*, who examined fish from a larger size range. In our study, jack mackerel $\delta^{13}$C was similar across the size gradient, but showed a significant but small (0.6‰) positive shift in $\delta^{15}$N from the smallest (43.2 g) to the largest (79.8 g) individual examined. In contrast to our results, *Espinoza et al. (2017)*, again in a wider size range, showed that jack mackerel $\delta^{15}$N was negatively related with individual size in the northern HCS. In order to have a reliable picture of ontogenetic shifts in $\delta^{13}$C and $\delta^{15}$N in N Chile, it is clear that larger samples are required both in terms of the number and of the size of individual fish.

Our estimates of anchovy consumption patterns and TP using stable isotope ratios are based on several key assumptions. For example, they are reliant on the use of representative values for the different prey groups and the isotopic baseline (phytoplankton) used to estimate TP, as well as for trophic discrimination (isotopic differences between prey and the consumer). With regard to our estimates of putative prey and the isotopic baseline, we followed a standardised sampling protocol across a considerable area (>2,000 km$^2$), which is likely representative for the wider region. Such wide-scale sampling should be sufficient to account for spatial differences in phytoplankton and prey $\delta^{15}$N associated with variation in in upwelling intensity, and hence the relative enrichment in $^{15}$N associated with this process (*Reddin et al., 2015*).

However, upwelling intensity (and the relative amount of $^{15}$N-enrichment) in the region also varies on a temporal scale (*Herrera & Escribano, 2006*), raising the possibility of a disconnect between phytoplankton (which rapidly reflect isotopic changes) and anchovies, whose tissue $\delta^{15}$N values may reflect conditions in an earlier time period, e.g., when phytoplankton $\delta^{15}$N was different than during the study period. Such isotopic decoupling has been reported elsewhere (*O'Reilly et al., 2002*) and, if not identified, can potentially confuse estimates of fish TP. We do not think this is a major issue in our study as the presence of favourable winds year-round means than coastal upwelling occurs throughout the year in the study region (*Palma, Escribano & Rosales, 2006*; *Thiel et al., 2007*). This suggest that there is unlikely to be a marked seasonal shift in $\delta^{15}$N values at the base of the food web that could lead to us underestimating baseline $\delta^{15}$N. Furthermore, anchovy growth is rapid in the study region, and continues throughout the year (*Cerna & Plaza, 2016*), lowering the risk of seasonal decoupling from a dynamic isotopic baseline.

In terms of TDF, we used a commonly applied $\delta^{15}$N TDF of 3.4 ± 1.0‰ to allow direct comparisons with *Espinoza et al. (2017)*. This value represents a mean estimated from

many different habitats, feeding strategies and taxa (*Post, 2002*). Studies of carnivorous fish often use a smaller $\delta^{15}$N value of 2.9 $\pm$ 0.3‰, following *McCutchan et al. (2003)*. Use of these values would increase our estimates of anchovy TP. Beyond the actual mean values used, it is also important to note that our estimates of TP were calculated following a robust Bayesian approach that includes error in both diet-tissue TDF and baseline $\delta^{15}$N (*Quezada-Romegialli et al., 2018*). Our approach however relies on the assumption that TDFs are additive, and this has been criticised by some authors. *Caut, Angulo & Courchamp (2009)* suggested that TDFs decrease with greater concentrations of $^{15}$N in the diet, and *Hussey et al. (2014)* developed a model to estimate what they refer to as scaled-TP. Their model includes a threshold (named $\delta^{15}N_{lim}$) estimating the situation where $\delta^{15}$N in the diet is such that trophic discrimination is expected to be zero. $\delta^{15}N_{lim}$ is calculated based on estimates for the slope ($\beta_0$) and the intercept ($\beta_1$) of a negative relationship between consumer TDF and the $\delta^{15}$N of their food based on experimental feeding studies. Using the Bayesian median estimates provided by Hussey and colleagues based on a meta-analysis, $\delta^{15}N_{lim}$ has a value of 21.9‰. Many consumers in the HCS of N Chile including benthic rockfish (*Docmac et al., 2017*) and the pelagic fishes detailed here (jack mackerel and high $\delta^{15}$N anchovies) have $\delta^{15}$N values greater than this threshold. This not only prevents the calculation of scaled TP estimates in these naturally $^{15}$N-enriched consumers, but highlights that the relationship between experimentally-derived TDF and dietary $\delta^{15}$N at the heart of the scaled-TP approach does not extend to natural ecosystems where $\delta^{15}$N is naturally high. Although this does not discount the Hussey et al. (*2014*) method, it does highlight that it is not universally applicable, and raises the need for more experimental studies.

Our mixing model results reflect an attempt to estimate the relative contribution of seven different putative prey groups using only two different isotopes. Although Bayesian mixing models can provide useful results where the number of potential sources is greater than the number of markers used (*Phillips et al., 2014*), the performance of our model might have been limited by both the large number of prey groups and the fact that some of our prey groups were isotopically similar (See Fig. 2), even though these groups differed taxonomically (and functionally). We attempted to counter this by including information on prey elemental concentration in the model (i.e., we used a concentration-dependent model) which will likely be most useful in distinguishing between animal and phytoplankton prey (*Phillips et al., 2014*). Given these caveats, we feel confident that our mixing model results (Fig. 4), support our conclusions that 1. anchovy are not assimilating significant amounts of C and N from phytoplankton; 2. that the anchovy population includes individuals feeding on two broad groups of animal prey, i.e., consumption of crustacean larvae by the low $\delta^{15}$N anchovy group, and ichthyoplankton (eggs and preflexion-stage anchovy larvae) by the high $\delta^{15}$N anchovy group (as well as juvenile jack mackerel).

Our data indicate that anchovy diet is both more variable than is generally considered and that some individuals can feed at high trophic positions. It has become increasingly clear that populations of consumers can include individuals following specialised trophic strategies (*Bolnick et al., 2003*). The inclusion of different trophic specialisms within anchovy populations may contribute to their productivity by limiting intraspecific competition

and support the unusually high fish production associated with the HCS (*Bakun & Weeks, 2008*; *Chavez & Messié, 2009*). The cannibalism and predation of eggs and larvae we suggest for the high $\delta^{15}$N anchovy group is commonly reported in clupeiform fishes (*Alheit, 1987*; *Hunter, 1981*), allowing consumption of an energy-rich prey (*Konchina, 1991*). The study area is recognised as a spawning and nursery area for fish, particularly anchovies (*Palma, Escribano & Rosales, 2006*; *Palma, Pizarro & Flores, 1992*). Given that anchovies and other fishes can spawn throughout the year in the region, this potentially results in a permanent supply of early life stage prey to predators, but may represent an important driver of natural mortality, with potential implications for future recruitment success.

## CONCLUSIONS

We used a stable isotope approach to examine the trophic ecology of one of the world's most important (but apparently understudied) fishes in an extremely productive, coastal upwelling zone. Our work has revealed several important findings with implications for how we understand and manage this important resource. We have shown that, like previous studies using stable isotopes from both central Chile (*Hückstadt, Rojas & Antezana, 2007*) and Peru (*Espinoza et al., 2017*), that anchovy TP has likely been considerably underestimated. Our estimates of anchovy TP differ from the classic value (2.2) used in much of the fisheries literature (*Guénette, Christensen & Pauly, 2008*; *Pauly et al., 1998*), and support *Espinoza et al.*'s (*2017*) observation that the classical hypothesis that short and efficient food chains drive secondary production in this ecosystem (*Ryther, 1969*) is no longer valid. We also have shown that the trophic ecology of individual anchovies is far more complicated than previously considered—individuals captured in the same net, fell into two broad trophic groups, even though they were largely of a similar size. These groups differed in trophic position and estimated diet, revealing previously unrecognised trophic variation in a fish that has long been considered to indiscriminately consume phytoplankton (i.e., TP~2.2). Given the similarities with the data presented by *Espinoza et al. (2017)* from anchovies further north in their distribution (which also showed considerable variation in $\delta^{15}$N), it is clear that any model assuming that anchovy are feeding at a low TP is suspect, and management decisions based on such models should be reconsidered.

Our use of stable isotope analysis to characterise ecosystem function in the pelagic zone of the HCS has shown that the system is more complex than previously thought. This parallels a recent study where, using a similar approach, we challenged long-held assumptions regarding energy supply in coastal kelp forest ecosystems in the north of Chile (*Docmac et al., 2017*). The current study highlights the utility of stable isotope analysis as a means to reveal cryptic ecological variation in systems that are generally considered well described. We are currently examining amino acid $\delta^{15}$N of different pelagic fishes from northern Chile including anchovy and jack mackerel to examine whether we see similar estimates of TP (*McClelland & Montoya, 2002*) and levels of individual variation as seen here with isotope analysis of bulk materials.

## ACKNOWLEDGEMENTS

We thank the handling editor and three different reviewers for useful comments on an earlier version of the manuscript.

### Funding

The study was supported by the Project FIP 2007-45 ''Efectos de la variabilidad de la capa de mínimo de oxígeno en la distribución y la abundancia de los principales recursos pesqueros de la zona norte''. Chris Harrod is supported by Nucleo Milenio INVASAL funded by Chile's government program, Iniciativa Científica Milenio from the Ministerio de Economía, Fomento y Turismo. Felipe Docmac is supported by the Rector of the Universidad de Antofagasta. Jessica Pizarro is supported by the Vicerrectoría de Investigación, Innovación y Postgrado of the Universidad Arturo Prat. The funders had no role in study design, data collection and analysis, decision to publish, or preparation of the manuscript.

### Grant Disclosures

The following grant information was disclosed by the authors:
Efectos de la variabilidad de la capa de mínimo de oxígeno en la distribución y la abundancia de los principales recursos pesqueros de la zona norte: Project FIP 2007-45.
Chile's government program, Iniciativa Científica Milenio from the Ministerio de Economía, Fomento y Turismo.
Vicerrectoría de Investigación, Innovación y Postgrado of the Universidad Arturo Prat.

### Competing Interests

The authors declare there are no competing interests.

### Author Contributions

- Jessica Pizarro conceived and designed the experiments, performed the experiments, analyzed the data, contributed reagents/materials/analysis tools, prepared figures and/or tables, authored or reviewed drafts of the paper, approved the final draft.
- Felipe Docmac analyzed the data, contributed reagents/materials/analysis tools, prepared figures and/or tables, authored or reviewed drafts of the paper, approved the final draft.
- Chris Harrod analyzed the data, contributed reagents/materials/analysis tools, prepared figures and/or tables, authored or reviewed drafts of the paper, approved the final draft.

### Animal Ethics

The following information was supplied relating to ethical approvals (i.e., approving body and any reference numbers):

Permission to sample was provided by the Chilean Subsecretaría de Pesca (Resolución Exenta N° 2205, 21 August, 2008) and by the Chilean Navy's Hydrographic and Oceanographic Service (SHOA N° 13270/24/317/VRS, 21 June, 2008).

## Data Availability

Stable isotope data and associated code for SIMMR mixing models and tRophicPostion are available in the Supplemental File.

## Supplemental Information

Supplemental information for this article can be found online at http://dx.doi.org/10.7717/peerj.6968#supplemental-information.

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
