# Peer review of "Clarifying a trophic black box: stable isotope analysis reveals unexpected dietary variation in the Peruvian anchovy Engraulis ringens"

_PeerJ, doi:10.7717/peerj.6968_

## Round 0.1 · original submission · Major Revisions

All 3 reviewers were generally enthusiastic about your study and provide constructive feedback about points to address when revising the manuscript. Although I have selected “major revisions” I believe the revisions required are probably more “moderate” than “major”. In addition, I have two review-level comments of my own. First, in the isotope methods section (lines 220-226) please provide the names of the standards that were used to normalize the carbon and nitrogen isotope data. Second, given that you have data from 2 isotopes, but 7 potential prey groups (and some with overlapping isotope values, particularly for δ13C, but also for δ15N), please comment in the manuscript about the robustness of the mixing model results (and associated interpretations) of dietary proportions for anchovy and jack mackerel that are shown in Figure 4.

Reviewer 1 ·

Basic reporting

Present study determined TP in Peruvian anchovy applying SIA method associated statistical approach. Large range of nitrogen isotope ratio within anchovies was resulted in the large variation of TP within species. I agree this interesting result and reasonable approach (e.g., mixing model and TP estimation) is acceptable, but need more concern for TP estimation and discussion for different diet preference within anchovy groups.

Experimental design

Present study applied “tROPHIC POSITION” for TP estimation, which is a new statistical approach. This approach looks nice to demonstrate TP of consumers by range scale. However, I think author need to explain more details about this new method. For example, nitrogen isotope ratio of phytoplankton was assumed as a baseline in this ecosystem. But nitrogen isotope value of phytoplankton has large range in this study. Thus I am wondering if SD of nitrogen isotope ratio of phytoplankton also included in the “tROPHIC POSITION” or only average value is used for baseline.
In addition, this study mentions that nitrogen isotope ratio of zooplankton is difficult because of uncertainty of TP. This assumption is reasonable, but need to concern relatively short turnover rate of phytoplankton than fish. In particular, study area is strongly affeceted by upwelling and hence variable nitrogen isotope baseline.

Validity of the findings

1. Author mentioned spatial difference in the carbon and nitrogen isotope ratio due to the upwelling intensity with previous studies, indicating large variability of isotopic baseline.
Similar isotope ratio of carbon among anchovies is suggested as an evidence for identical nitrogen isotopic baseline, due to the similar pattern of spatial variation between carbon and nitrogen isotopes reported in the previous studies. However, this evidence are not enough, and also unclear. Present study collected samples for diet resources in 16 sites with large range of isotope ratios. This can be demonstrated spatial variation of carbon and nitrogen. This is maybe more helpful to the readers.
Also, author mentioned that sampling area is recognized as a spawning and nursery area. If anchovy migrate from other habitat, their isotope might be recorded information of the former habitat, meaning different nitrogen/carbon isotopic baseline.
Therefore, more information of the migration scale for anchovies is required.

2. Based on TP, author identified two groups of anchovies, indicating different diet preference. However, in my idea, assuming large acceptability of diet in anchovies, TP of anchovy may be distributed continuously large range rather than grouping. Author should discuss the possible reason for this grouping in anchovies.

3. In many part of this MS, author emphasizes higher TP than 2 in anchovy (previous knowledge) as new finding in this study. However, in my feeling, this is not interesting and weak for main discussion. This result clearly showed different feeding strategy between anchovy and jack mackerel, which may imply importance role of these fish in ecological view. I recommend more deep discussion (or reason) for how and why anchovy have large range of TP.

Additional comments

Because I am not a native in English, I can judge English in this MS. However, I found several mistakes and strange English in the MS, which should be revised before published. For example,
- Author use both trophic level and trophic position. I recommend identical term.

- Line 400 Docmac et al, 2016: cannot find in the reference lists (Docmac et al. 2017 ?)

- For carbon isotope value author did not use minus sign “−”

- Line 239 npermutations = 9 999 : need to correct
- Line 341 Espinoza et al’s : when?

Reviewer 2 ·

Basic reporting

Overall, this was a very well-written stable isotope study focused on the trophic ecology of the Peruvian anchovy. Aside from a few grammatical errors, the language was quite clear with excellent English. The Intro, while probably a bit longer than necessary was thorough and appropriate references were included. The Figures were clear and well-constructed. I might suggest stacking panels A & B of figure 3 so that a single x-axis can be used for both panels but that is a minor comment and more stylistic than anything else. The use of horizontal lines throughout Table 1 is a little jarring and not typical of most journal formats so I point it out but leave it to the copy editors to determine if that is an issue. The data were provided in zip format with the ms.

Experimental design

The authors’ findings that stable isotope-based estimates of trophic position of the fish were markedly higher than estimates from stomach contents (in agreement with several other recent studies) were based on sound, well-established methods (although I note one consideration that should be addressed below under General comments). The study falls within the scope of PeerJ and focuses on an ecologically important issue. In several instances, noted below, the methods were not entirely clear. Assuming the authors provide clarification, description of the methods is/will be quite clear and replicable.

Validity of the findings

The data appear to support the authors’ contentions, the conclusions are reasonable and the Discussion is appropriate.

Additional comments

General comments
Overall, I think this is a very nice study that I enjoyed reviewing. I have only one major comment (follows) and a number of specific comments (below).

The only concern I have is the potential for seasonality in the phyto/zooplankton d15N values to bias estimates of fish trophic position due to seasonal changes in the strength of upwelling. The authors state that fish and putative prey were collected during the austral winter but that upwelling in the region is typically strongest during the austral spring-summer (Line 172-173). Upwelling brings more isotopically heavy, deep water NO3 to the surface and it’s quite possible that could result in heavier phytoplankton and zooplankton d15N values during the spring-summer (e.g., as discussed in Hannides et al., 2009. Quantification of zooplankton trophic position in the North Pacific Subtropical Gyre using stable nitrogen isotopes). Also, spring-summer seasons in many areas are associated with higher somatic growth rates of fish and therefore can have a stronger influence on the stable isotope values of fish muscle tissue (e.g., Perga and Gerdeaux, 2005. ‘Are fish what they eat' all year round?). If the anchovies collected during the winter have tissues that are more reflective of summer conditions when prey had heavier d15N values, then the net result would be an overestimate of trophic position when using lighter winter d15N values as a baseline. Since the higher than expected TP values of anchovies estimated here are a key take-home message from this paper, I’d like the authors to address this point. I don’t think this potential issue invalidates the current study, but I think it is a consideration that needs to be stated for readers, discussed and noted as deserving future research. If, on the other hand, there is previous literature from the study region that demonstrates no seasonality in stable isotope values of phyto/zooplankton linked to changes in the strength of upwelling, then the authors should definitely make this known to readers and cite the appropriate literature.

Specific comments
Line 96 – upwelling systems are generally considered very productive for pelagic fisheries (as noted by the authors) and the HCS is no exception, therefore I would recommend deleting ‘anomalous’ from this sentence
Line 107 – add ‘in’ after ‘role’
Line 109-110 – presumably the HCS anchovy stats skew the global mean TL of catch downward….
Line 118 – do you mean ‘HCS’?
Line 117-121 – I know what the authors are arguing here, but ‘changes in our understanding’ of diet doesn’t actually impact ‘the movement of energy and nutrients through the food web’. Please adjust the wording of this sentence to reflect how our knowledge can influence our understanding of food web structure rather than the actual structure itself.
Line 145 – delete ‘is’
Lines 156-166 – These paragraphs can be integrated
Line 203-205 – lipid removal is considered standard practice for high-lipid fish tissues but it is not considered necessary for low-lipid tissues. If available, please provide a measure of lipid content of tissue or accepted proxy (e.g., C:N ratio) from pre-extracted tissue so readers can evaluate the need for lipid-removal
Line 204 – oven dried or freeze-dried? If oven, please note temperature
Line 213-216 – this section is a little unclear. Where phytoplankton really ‘placed’ on filters or were they filtered onto the GFFs? For zooplankton, it sounds like they were dried on GFFs, acidified, then scraped off the GFFs, homogenized, then analyzed…is this correct? How were samples acidified? It sounds like they were fumigated (if so, please make this explicit in the text), but if acid was directly applied, then please note that fact.
Line 216 – change ‘sampled’ to ‘samples’
Line 249-250 – Presumably, literature diet studies were used to decide which groups of plankton to use as potential prey from the range of taxa identified (Lines 208-211). Please provide 1 or more suitable references
Line 268 – change ‘mas’ to ‘mass’
Line 268 – delete extra set of parentheses around the SD of mass
Line 274 – here and elsewhere, a single notice that values following the +/- sign are SDs is sufficient. It shouldn’t be necessary to repeat this notice for each usage
Line 274 – this is a HUGE range in dN for one species!
Lines 299-306 – I’m confused as to why the authors elected to use PERMANOVA for these two univariate statistical models. Why not just use ANOVA or GLMs (if you want to include covariates) to look at group-based differences in dN or dC? Perhaps I’m missing something?
Line 348 – a word is missing in this sentence
Line 349 – Can you add ‘naturally’ before ‘labelled’ to emphasize that these observations come from naturally occurring heavy isotope values in deep ocean water rather than experimental labelling with heavy dN?
Line 353-356 – could spatial differences in phyto assemblage composition also contribute to the observed variability?
Lines 406-409 – It is also important (more so?) to consider body length relative to TP when looking for ontogenetic shifts b/c body length is more likely to positively covary with individual gape size, which is a very important factor influencing the types and sizes of potential prey as fish grow. The overall length range of fish analyzed here was quite narrow and presumably this is why the authors focused on mass which had a wider total range.
Line 417 – I think the authors meant to use ‘synoptic’ rather than ‘symptomatic’
Line 418 – please translate square NM to square km to remain consistent with use of metric in the ms

·

Basic reporting

No comment

Experimental design

In the section Material & methods:

First observation:
Sample collection:
On the second paragraph, the authors do not give details of hour collection.
This factor is important to know, taking into account some reports about light influence on fractionation at inverse relationship.
One of these reports is:
Needoba, JAN, Waser, NA, Harrison, PJ, Calvert, SE. (2003). Nitrogen isotope fractionation in 12 species of marine phytoplankton during growth on nitrate. Marine Ecology Progress Series 255, 81-91.

Second observation:
Data analysis:
On the first paragraph, is not so clear why use Pearson correlation index and Welch t-test?.
There are not information on previous analysis such as normality and homocedasticity test.

I advice follow the routine:
- test the normality of the data (may be with Shapiro-Wilk), if the data is normally distributed, follow the next step.
- test of homocedasticity (Levene test), if the variance are not equal follow the next step.
- Welch t-test to examine differences in the data when compare two variables.
- Finally, apply the Pearson correlation test, which is parametric.

The other way is use the non-parametric tests, and Spearman correlation test is adviced.

Validity of the findings

No comment

---

## Round 0.2 · Minor Revisions

The authors have done a thorough job revising their manuscript and addressing the comments and suggestions of the reviewers. There is one remaining point that I would like the authors to address: in my feedback on the original version of their manuscript I asked them to provide the names of the standards used to normalize the carbon and nitrogen isotope data. They responded by reporting the precision associated with repeated analyses of an internal standard, "peach leaves". I appreciate this added detail. However, what I was hoping the authors would provide was more information about how the raw isotope data from the IRMS were normalized to the international scales. For example, was a single- or two (or more)-point normalization curve used? Were internal (lab) or international standards used to create this curve? If internal standards were used, then what international standards can the isotopic values of those internal standards be traced back to? In my opinion, such information is under reported in the stable isotope literature, but is important for making our science as rigorous and reproducible as possible. Here is an example of how I typically report such information: "The δ13C and δ15N data were normalized to the VPDB and AIR scales, respectively, using a two-point normalization curve with laboratory standards calibrated against USGS40 and USGS41." I hope this helps to clarify my original comment. I am happy to discuss further if needed.

---

## Round 0.3 · accepted · Accept

Thank you for adding the minor, but important, methodological details I requested.

#